Landscape fragmentation and pollinator movement within agricultural environments: a modelling framework for exploring foraging and movement ecology

Rands Sean A. sean.rands@bristol.ac.uk
School of Biological Sciences, University of Bristol , Bristol , United Kingdom
de Vere Natasha
Electronic publication date: 2014 Feb 27
Publication date: 2014
Volume: 2
Electronic Location ID: e269
Received 2013 Apr 17; Accepted 2014 Jan 21
Copyright: © 2014 Rands
Copyright year: 2014
Copyright holder: Rands
License: This is an open access article distributed under the terms of the Creative Commons Attribution License, which permits unrestricted use, distribution, reproduction and adaptation in any medium and for any purpose provided that it is properly attributed. For attribution, the original author(s), title, publication source (PeerJ) and either DOI or URL of the article must be cited.
License URL: https://creativecommons.org/licenses/by/4.0/

Keywords: Bumblebee, Honeybee, Behavioural ecology, Movement ecology, Simulation, Random walk, Foraging distance, Behavioural rules, Pollinator biology, Ecosystem services

Funding: This work was unfunded.

==============================
Pollinator decline has been linked to landscape change, through both habitat fragmentation and the loss of habitat suitable for the pollinators to live within. One method for exploring why landscape change should affect pollinator populations is to combine individual-level behavioural ecological techniques with larger-scale landscape ecology. A modelling framework is described that uses spatially-explicit individual-based models to explore the effects of individual behavioural rules within a landscape. The technique described gives a simple method for exploring the effects of the removal of wild corridors, and the creation of wild set-aside fields: interventions that are common to many national agricultural policies. The effects of these manipulations on central-place nesting pollinators are varied, and depend upon the behavioural rules that the pollinators are using to move through the environment. The value of this modelling framework is discussed, and future directions for exploration are identified.

Introduction

Pollinators provide vital ecosystem services in both wild habitats and the agricultural landscape, where they contribute enormous economic value to the production of crop species (Morandin & Winston, 2006; Gallai et al., 2009). In addition to managed pollination by honeybees, it is being increasingly acknowledged that unmanaged wild species may be providing a vast amount of pollination within the managed environment (Winfree et al., 2007; Holzschuh, Dudenhöffer & Tscharntke, 2012; Garibaldi et al., 2013). However, it is acknowledged that many species of pollinator are in decline (Biesmeijer et al., 2006; Goulson, Lye & Darvill, 2008). Considerable scientific effort is currently being devoted to understanding this decline (Potts et al., 2010), in an effort to identify strategies for both arresting and reversing it (Brown & Paxton, 2009; Winfree, 2010).

Land-use change is frequently considered to be a major contributor to pollinator decline (Potts et al., 2010), through both habitat fragmentation (but see Hadley & Betts, 2012) and the loss of habitat suitable for pollinators (Fischer & Lindenmayer, 2007). Many regional agricultural management schemes aim to counteract this, through the creation of wild habitat within the agricultural landscape. This is presumed to be beneficial to pollinators, through providing either wild refuge areas, or providing corridors to facilitate movement (e.g., Batáry et al., 2011; Ernoult et al., in press). These landscape manipulations may therefore be beneficial to enhancing pollination within the landscape. Patches of wild refuge and set-aside land have been demonstrated to enhance biodiversity with the agricultural environment (van Buskirk & Willi, 2004; Orłowski, 2010; Carvalheiro et al., 2011; Norfolk et al., 2013). However, the value of corridors is debatable: corridors can both aid pollinators to move through the environment (such as through giving visual signposting or an obstruction-free route) and hinder their movement (such as by providing physical barriers) through the environment (Collinge, 2000; Fried, Levey & Hogsette, 2005; Davies & Pullin, 2007; Beier & Noss, 2008; Öckinger & Smith, 2008), and may even be harmful if they allow the spread of invasive species (Procheş et al., 2005). Even if a corridor is demonstrated to be a useful feature to add to the environment, the corridor on its own may not provide extra value to the landscape, as the composition of the landscape adjacent to the corridors may also contribute to how well they function (Baum et al., 2004).

Because the evidence is relatively mixed for the value of these mitigation strategies, we therefore need to better understand the effects that these different environmental manipulations have on the pollinators that live within them. As well as observational studies comparing existing manipulations, we can conduct experimental manipulations (Jenerette & Shen, 2012). We can also investigate the biology and effects of the manipulations using theoretical models, which allow us to explore many different scenarios without conducting expensive and time-consuming field trials. Careful model formulation allows us to identify aspects of the biology of the pollinating species that may impact on how they interact with the environment. In particular, as urged by Lima & Zollner (1996), we can tie concepts from behavioural ecology with landscape ecology, to better inform how organisms are interacting with the habitat in which they live.

Techniques such as resource-selection function models use characteristics of the environment and the physiology and potential nesting locations of pollinating species of interest, and have been used to identify the likely site of foraging (Forester, Im & Rathouz, 2009; Lonsdorf et al., 2009; Henry et al., 2012). However, further realism can come if we tie spatially-explicit landscapes with well-developed concepts from behavioural ecology. Individual-based models of movement within a landscape allow us to model the movement path of individuals, based on their internal state (e.g., physiological requirements) and their capacity to perceive and move through the environment, and can accommodate external factors such as structure of the environment (Nathan et al., 2008; Martin et al., 2013). Spatially-explicit individual-based models (McLane et al., 2011) have been used to explore diverse questions in behavioural ecology (e.g., Rands et al., 2004; Rands et al., 2006; Rands, 2012). We would expect that the foraging decisions and movements of pollinators will be affected by local resource availability, resource quality, and the ease of locating resources and moving through the environment. In turn, these can be tied to the physical composition of the landscape. All of these factors will be changing dynamically, and will be subject to weather, interference from other foragers, and anthropogenic change within the environment. Spatially-explicit individual-based techniques are therefore ideal for exploring the effects of habitat fragmentation and change on the behaviour of pollinators nesting within the environment, as they allow us to consider the effects of behavioural rules within a spatially complex environment.

Here, I develop a framework for considering pollinator movement within the environment, using a spatially-explicit individual-based model of the behaviour of a central-place forager that is nesting within its environment. I build on the spatially-explicit models presented by Rands & Whitney (2010) and Rands & Whitney (2011), which simply considered the proportion of resources available within a maximum foraging range to a nest, and the effects that the ratio of resources had upon mean uptake rates experienced by the pollinators. Although they considered the effects of behavioural preferences on intake, these initial models did not consider how pollinators moved through the environment. Considering behavioural movement rules is important, but this consideration is frequently missing from landscape ecology models (Bélisle, 2005). In this framework, I consider a landscape that can be seen as a mixture of foraging opportunities, rather than as a series of connected patches. The structure of the landscape will affect how a pollinator experiences foraging sites, and it is possible that specific behavioural rules will mean that the pollinator is less likely to reach some areas of the landscape if their movement rules do not allow them to reach that area. Understanding these local effects is important, because many important pollinators do not travel far (Zurbuchen et al., 2009), and are constrained to remain near their nest. I develop a framework for an agricultural field system where fields are surrounded by strips of wild land, and consider how pollinators following some simple behavioural rules might move through this environment. I then explore whether landscape manipulations (removing corridors, and adding ‘set-aside’ wild land) have effects upon the amount of movement and the behavioural choices made by foraging pollinators within the landscape. I also consider the special case for specialist pollinators that are unable to forage outside the wild regions of the environment, and consider how landscape manipulation can affect their movement.

Methods

The field-based landscape within these simulations is initially generated as a grid-based Voronoi tessellation, described in detail in Rands & Whitney (2011), where an initial grid of square cells is seeded with a given number of field seeds. Fields are then calculated by allocating individual cells in the grid to the field characterised by the seed closest to that cell. Field edges, considered as ‘wild’ land, are cells that have at least one immediate neighbour that belongs to a different field, following Rands & Whitney (2011). Any part of the environment that is not a field edge is considered to be a field, and contains cultivated land (considered as monoculture, following Rands & Whitney, 2010; Rands & Whitney, 2011). This means that all the fields of cultivated land are separated from each other by field edges, which could be considered as hedgerows, or wild strips, or any other form of non-cultivated component of the landscape. The nests of the pollinators (I am assuming these to be a central-place foraging pollinator that returns to a nest, such as solitary bee or bumblebee) are assumed to be located within the field edges, as the cultivated component of the landscape may be too disturbed to allow a nest to survive. The (x, y) coordinates of the nest are taken to be the initial starting position of a foraging pollinator.

The state of an individual foraging pollinator is defined by its position (in (x, y) coordinates) and its current direction of travel, which is equal to the direction it moved in from the previous to current square. An individual can move in any of the four directions available by moving to a grid square sharing a side with the current position. The neighbouring square that would be entered if the pollinator continued in its current direction of travel is defined as the ‘forward’ square, and the pollinator is defined as travelling ‘forward’ into it. The neighbouring square if the pollinator rotated 90° clockwise from its current direction of travel is the ‘right’ square (and the pollinator travels ‘right’). Similarly, the squares that would be entered if the pollinator rotated 180° and 270° clockwise are the ‘backwards’ and ‘left’ squares respectively (with the pollinator travelling ‘backwards’ and ‘left’).

I assume movement follows a correlated random walk. If we initially ignore the contents of the grid, the unadjusted probabilities that the pollinator moves either forwards or backwards are pF and pB. I assume that the pollinator’s unadjusted tendency to deviate from going forwards is symmetrical, so the chances of moving to the squares on the left or right sides have equal probabilities, both pS, where pF + pB + 2pS = 1. However, the probabilities that the pollinator moves into neighbouring squares is also influenced by the pollinator’s tendency to switch between habitat types. I assume that the content of its current location is ccurrent, and the contents of the neighbouring squares forward, backward, left and right of the current square are cF, cB, cL and cR respectively. I then assume an adjusted preference mi for entering square i where switching habitat type incurs a reduction r, such that mi=piif ci=ccurrent,rpiif ci≠ccurrent.

The actual probability ai that an individual moves in direction i is calculated as ai = mi/(mF + mB + mL + mR). Using these four probabilities of movement, the pollinator then randomly picks its direction of movement for the period.

All simulations were written in C++ (available in Supplemental Information 1) and run using XCode 4 and 5 (Apple, Inc., Cupertino). Random numbers were generated within the simulations using a Mersenne twister algorithm (Matsumoto & Nishimura, 1998).

Model 1: effects of movement choice

1000 environments were independently generated. Each environment was 1000 × 1000 uniform squares with an edge length of one grid unit, and had a random number of fields (chosen from the range [101, 200]) seeded within the environment using randomly chosen coordinates. Voronoi-tessellated field boundaries were then created using the algorithm described in Rands & Whitney (2011), assuming that field edges were of a single thickness: this meant that every cell in a hedgerow was connected via at least one of its edges to another hedgerow cell (see Fig. 1 for an example).

Figure 1 Illustration of how set-asides were added into the landscape.

The left hand panel shows a 101 × 101 cell landscape generated using 30 randomly placed field seeds, where white cells represent agricultural crops and black cells represent wild land or hedgerows. Set-aside fields are added by randomly selecting fields containing agricultural crops, and resetting the cells within the field as wild land. Moving from left to right, each successive panel has two additional agricultural fields redesignated as set-aside. Note that this is a simplified sketch: the results described consider a larger landscape and add more than two fields at each assay point.

For each environment, a switching reduction r was randomly selected from (0, 1), and a random constant v was randomly selected from (0, 1). Single model runs were then calculated using the environment field description, each with systematic alteration of pF where pF∈{0.25, 0.375, …, 0.875}, pS = ((1−pF)/3) + (v(1−pF)/6), and pB = 1−pF−2pS. In each model run, a pollinator that started at the nest was followed for 1000 timesteps. Over these timesteps, I recorded the maximum distance the pollinator travelled from the nest, the number of times it switched between habitat types, and the proportion of time it spent in the hedgerow habitat type. These calculations were repeated over all of the 1000 environment types generated.

Model 2: effects of altering the probability of switching foraging habitat

The modelling of the environment was similar to Model 1. However, rather than systematically altering pF, r was systematically altered, where r∈{(2/3)0, (2/3)1, …, (2/3)13}. For each environment, pF was randomly chosen from (0.25, 1), pS = ((1−pF)/3) + (v(1−pF)/6) where v was randomly chosen from (0, 1), and pB = 1−pF−2pS.

Model 3: effects of including set-asides

The modelling of the environment was similar to Model 2, but within each environment, r was randomly chosen value within (0, 1). For each environment, a basal landscape of fields with hedgerows was created as described in Model 1, and pollinator movement statistics were calculated. Five of the fields were then randomly selected as set-asides, and all the squares within these set-aside fields were filled with wild hedgerow vegetation (see Fig. 1 for a sketch of how this was implemented), and pollinator movement statistics were calculated. Five more fields were then filled as set-asides (giving the environment ten set-asides in total), with movement statistics calculated. This addition of five set-aside fields with movement calculations was repeated until fifty of the original fields had been filled as set-asides.

Factorial sensitivity analysis (Hamby, 1994) was also conducted, by systematically increasing the number of set-asides over the range (0, 5, …, 50) whilst varying (a) pF over (0.1, 0.2, …, 0.9); (b) pS over (0, 0.05, …, 0.45) whilst setting pF = (1−2pS) × (random value from range [0,1]); and (c) r over (0, 0.1, …, 0.9). 1000 different randomised sets of the parameters that were not being investigated within each set were generated as described in the main Methods section. For each of these parameter sets, one individual pollinator was moved through the generated environment for all possible combinations of the pair of parameters that were being investigated.

Model 3a: effects of including set-asides when pollinators do not change habitat

This model was identical to Model 3, but r = 0, meaning that the pollinators did not swap habitats, and therefore did not move into fields that were not set-aside. Only the maximum geometric distance from the nest was calculated.

Model 4: effects of removing hedgerows

The modelling of the environment and calculation of movement was calculated in a similar way to Model 3. However, rather than filling fields as set-asides, the basal environments were altered by cumulatively removing the hedgerows between fields. An individual hedge was considered to be the grid squares designated as hedgerow that fall between two identifiable field seeds, similar to a vertex in a Voronoi tesselation. Movement statistics were calculated for the basal environment and then after every four consecutive hedgerow removals, meaning that movement statistics were calculated after 0, 4, …, 40 hedges were removed (see Fig. 2 for a sketch of how this was implemented). Factorial sensitivity analysis (Hamby, 1994) for the effects of pF, pS and r on Model 4 were conducted in a similar way to those described above for Model 3, but by systematically increasing the number of hedgerows removed over the range (0, 4, …, 40) rather than the number of set-asides present.

Figure 2 Illustration of how hedgerows were removed from the landscape.

The left hand panel shows a 101 × 101 cell landscape generated using 30 randomly placed field seeds, where white cells represent agricultural crops and black cells represent wild land or hedgerows. Hedgerows are removed by randomly selecting adjacent fields, and removing the cells between them that were initially designated as hedgerows. Moving from left to right, each successive panel has four additional hedgerows removed.

Model 4a: effects of removing hedgerows when pollinators do not change habitat

This model was identical to Model 4, but r = 0, meaning that the pollinators did not swap habitats, and therefore could become more limited in their movements as the removal of hedgerows fragmented the corridors within the landscape. Only the maximum geometric distance from the nest was calculated.

Statistical analysis

Using lme4 0.999999-0 (Bates, Maechler & Bolker, 2012) in R 2.15.1 (R Development Core Team, 2012), the three measures of movement (maximum distance from nest, number of habitat changes, and amount of time in wild habitat) were modelled separately as linear mixed models against the variable being systematically changed in each model, with simulation run (with a specific fixed set of randomly generated environmental parameters) considered to be a random factor. Models including the systematically altered variable were compared with the equivalent null models missing the variable, and these were compared using a likelihood ratio test. If this test was significant, post-hoc pairwise Tukey tests were conducted using multcomp 1.2–12 (Hothorn, Bretz & Westfal, 2012) to explore the shape of the relationship.

Results

Model 1: effects of movement choice

As would be expected, increasing the likelihood of choosing to move forward during a period (pF) increased the maximum distance away from the nest that an individual reached (Fig. 3A, Table 1). Although there were differences in the number of times the forager switched environment (Table 1), this trend had no obvious pattern (Fig. 3B, Supplemental Information 10). There was a tendency for the proportion of time spent in the wild environment to reduce as the probability of moving forwards increased (Table 1), but this reduction was small (Fig. 3C).

Figure 3 Box plots showing trends for Model 1.

Box plots show the effects of changing the probability of moving forwards in a period (pF) on the median value of: (A) maximum distance travelled away from the nest in 1000 movements; (B) the number of times the forager changes habitat; and (C) the proportion of time the forager spends in the ‘wild’ habitat.

Table 1 Overall changes in mean summary statistics of pollinator movement for the different models.

⇑⇑ / ⇓⇓: strong increase/decrease across the range of the parameter changed, with most post-hoc comparisons significant (see Supplemental Information 10); ⇑ / ⇓: moderate increase/decrease across the range of the parameter changed, with some post-hoc comparisons significant; (⇓): significant decrease in response to parameter being changed only significant at one extreme end of range considered; —: no obvious pattern in response to parameter being changed, regardless of significance (or lack of significance) in post-hoc comparisons. Statistics reported are for likelihood ratio tests.

	Maximum distance
from nest	Number of habitat
changes	Proportion of time
spent in wild habitat	
Model 1: increasing pF	⇑⇑	χ52=4768.9,p<0.001	—	χ52=14.4,p=0.013	⇓	χ52=91.1,p<0.001	
Model 2: decreasing r	⇑⇑	χ132=542.2,p<0.001	⇑⇑	χ52=10446.0,p<0.001	⇓⇓	χ132=4213.7,p<0.001	
Model 3: increasing set-aside number	(⇓)	χ102=71.5,p<0.001	⇓⇓	χ102=1225.0,p<0.001	⇑⇑	χ102=3154.0,p<0.001	
Model 3a: increasing set-aside number with no movement into cultivated fields	⇑⇑	χ102=1146.8,p<0.001					
Model 4: increasing number of hedgerows removed	—	χ102=56.5,p<0.001	⇓	χ102=90.5,p<0.001	(⇓)	χ102=52.5,p<0.001	
Model 4a: increasing number of hedgerows removed with no movement into cultivated fields	—	χ102=22.2,p=0.014					

Model 2: effects of altering the probability of switching foraging habitat

Increasing the probability of switching foraging habitat during a period (r) led to a small increase in the maximum distance travelled away from the nest (Fig. 4A, Table 1). As would be expected, increasing the probability of switching led to an increase in number of switches (Fig. 4B, Table 1). There was a reduction in the time spent in ‘wild’ habitat (Fig. 4C, Table 1): foragers were much more likely to spend time in the ‘wild’ habitat if they were not likely to switch habitat, presumably because they began their foraging trip within the ‘wild’ habitat.

Figure 4 Box plots showing trends for Model 2.

Box plots show the effects of changing the probability of switching foraging habitat during a period (r) on the median value of: (A) maximum distance travelled away from the nest in 1000 movements; (B) the number of times the forager changes habitat; and (C) the proportion of time the forager spends in the ‘wild’ habitat.

Model 3: effects of including set-asides

Increasing the number of set-aside fields within the habitat had little effect upon the maximum distance foragers travelled away from their nest (Fig. 5A, Table 1), but led to a decrease in the time they switched between habitats (Fig. 5B, Table 1) and an increase in the time they spent within the ‘wild’ habitat (Fig. 5C, Table 1). Sensitivity analyses are presented in Supplemental Information 2–5: trends for two parameters combined follow what is expected when each of the parameters are considered individually, and there appear to be no unexpected interactions between parameters.

Figure 5 Box plots showing trends for Models 3 (panels A–C) and 3a (panel D).

Box plots show the effects of changing the number of set-aside fields in the foraging environment on the median value of: (A) maximum distance travelled away from the nest in 1000 movements; (B) the number of times the forager changes habitat; (C) the proportion of time the forager spends in the ‘wild’ habitat; and (D) the maximum distance travelled away from the nest when the forager never crosses into the ‘non-wild’ habitat.

Model 3a: effects of including set-asides when pollinators do not change habitat

Increasing the number of set-aside fields within the habitat led to a slight increase in the distance travelled from the nest when the foragers were constrained to remain within the ‘wild’ habitat (Fig. 5D, Table 1).

Model 4: effects of removing hedgerows

Although there was an effect of increasing the number of hedgerows removed from the environment upon the maximum distance a forager travelled away from its nest (Table 1), this effect did not yield an easily describable trend (Fig. 6A, Supplemental Information 10). Increasing the number of hedgerows removed led to a slight decrease in the number of times the forager changed habitat (Fig. 6B, Table 1) and the time spent in the ‘wild’ habitat (Fig. 6C, Table 1). Sensitivity analyses are presented in Supplemental Information 6–9: trends for two parameters combined follow what is expected when each of the parameters are considered individually, and there appear to be no unexpected interactions between parameters.

Figure 6 Box plots showing trends for Models 4 (panels A–C) and 4a (panel D).

Box plots show the effects of changing the number of hedgerows removed from the foraging environment on the median value of: (A) maximum distance travelled away from the nest in 1000 movements; (B) the number of times the forager changes habitat; (C) the proportion of time the forager spends in the ‘wild’ habitat; and (D) the maximum distance travelled away from the nest when the forager never crosses into the ‘non-wild’ habitat.

Model 4a: effects of removing hedgerows when pollinators do not change habitat

As for Model 4, although there was an effect of removing hedgerows on the maximum distance a forager moved when it was constrained to stay within the ‘wild’ habitat (Table 1), the trend seen was not a simple increase or decrease (Fig. 6D, Supplemental Information 10).

Discussion

Movement rules

Movement rules are important in determining where and how far pollinators travel. In the first two models, I demonstrate this using two simple behavioural rules based on simple random walks and habitat preferences (where the habitat preference results echo those described by Rands & Whitney (2010)). Many other behavioural rules for moving through landscapes are possible (Getz & Saltz, 2008), and understanding the movement process itself is arguably a key consideration in formulating realistic and useful models of animal movement within the environment (Schick et al., 2008). However, I chose to use naïve directed random walks in this example for the sake of keeping this initial framework simple.

More realistic rules are likely to involve some degree of state-dependence, taking into account dynamic changes in both the external environment and the internal state of the moving individual (Rands et al., 2004; Rands et al., 2006; Nathan et al., 2008; Martin et al., 2013). Taking a behavioural ecology approach, ideally we want to identify a behavioural rule-set that optimises the fitness of an individual, based on how its actions are influenced by internal state and the environment (Houston & McNamara, 1999). Both of these may change dynamically in response to the actions conducted. Furthermore, we may also need to consider how the movement rules are constrained by the behavioural mechanisms that can be used (McNamara & Houston, 2009), which may not be able to exactly enact the exact optimal behaviour identified (Rands, 2011). The modelling framework discussed here uses a toy example of the movement behaviour used by individuals, but could be refined to consider an optimal rule-set (or indeed could be used to identify those rule-sets which allow individuals to respond appropriately to their current environment, such as shown by Morrell, Ruxton & James (2011)). If the framework was being used to identify optimal rules, it would be essential to identify an appropriate currency to optimise: see Rands & Whitney (2008) and Charlton & Houston (2010) for discussion of which currencies may be appropriate to central-place nesting pollinators.

To fully understand movement through the environment, we need to consider the behavioural rules shown by individuals at the small, local scale (such as within patches of flowers, where pollinators are choosing how they move between individual flowers) and at the larger landscape scale: the framework I present here is more suited to the latter of these. Much empirical and theoretical work has been devoted to understanding finer-scale local rules, and many experiments picking apart choice behaviour have been done within confined apparatus that may be constraining the rule set open to the pollinator. Many central-place foragers are known to trapline, where they form and then maintain an established route of visitation during a foraging bout, which may act to optimise the amount of resource they are returning to their nest (Lihoreau et al., 2012). This has been demonstrated to both enhance the resources gained by the forager (Ohashi & Thomson, 2009; Reynolds, Lihoreau & Chittka, 2013), and possibly enhance the pollen flow between the plants visited (Ohashi & Thomson, 2009). Memory and the requirement for environmental sampling should therefore also be considered within a realistic model of pollinator movement.

The effects of landscape manipulation

The framework presented gives a clean and simple method for exploring the effects of landscape manipulation on the movement behaviour of pollinators. Increasing the number of set-asides gave an increase in the amount of time spent in ‘wild’ habitat, and a decrease in the amount of habitat switching (Model 3). This is unsurprising, given that the amount of wild land in the immediate environment of the pollinator was being increased, with the accompanying increase in the average number of wild squares neighbouring a focal wild square. The effects of removing wild strips was slight (Model 4), and had little effect upon the maximum distance travelled from the nest.

Therefore, although the modelling technique was relatively simple, it is likely that the movement rules used were too simple to accurately reflect what might be happening when pollinators are responding to anthropogenic change. Given more suitable behavioural rules, the framework could be used to investigate the effects of field shape, as it has been demonstrated that field shape is a factor that can affect the presence of invertebrates in an agricultural environment (Yaacobi, Ziv & Rosenzweig, 2007; Orrock et al., 2011). It would be relatively easy to manipulate simplified raster-based landscape information (such as the UK mapping data used by Rands & Whitney (2011)) to explore the effects of landscape manipulation within a specific agricultural environment.

Landscape manipulation and specialist pollinators

I also considered the effects of environmental manipulations on the travelling distance of specialist pollinators that were unable to move into the cultivated landscape (Models 3a and 4a). Including set-aside wild fields slightly increased the distance travelled, but the removal of corridors had subtle and mixed effects. Arguably, the rules used in this model were too simple to characterise the behaviour of these pollinators: being constrained to forage on a particular species of plant does not mean that movement needs to be constrained to the areas in which those plants grow. Our movement rules therefore need to consider whether managed features of the landscape have similar effects in different species. For example, hedgerows have many differing physical and ecological effects on the agricultural landscape (Forman & Baudry, 1984). Although they are relatively undisturbed relative to the surrounding landscape (therefore potentially acting as wild refuge zones), they can also act as physical barriers to dispersal (Joyce, Holland & Doncaster, 1999; Wratten et al., 2003). Furthermore, if landscape structures are known to impede the movement of pollinators, we still need to be careful to ensure that a barrier is not labelled as impassable without good experimental evidence (Zurbuchen et al., 2010).

In the model, I characterise the environment as being composed of two environment types: agricultural crop, and wild flowers, and assume the pollinator is switching between the two, partially influenced by learned preferences and neophobia (Rands & Whitney, 2010; Rands & Whitney, 2011). Different floral species may be more or less attractive depending upon their spatial distribution and proximity (Nattero et al., 2011; Hanoteaux, Tielbörger & Seifan, 2013), and creating a spatially realistic model that accounts for the distribution of multiple forage types may require us to understand all the switches in preference that could occur when a pollinator is able to move from one forage type to another. This may in turn require a lot of careful experimentation considering all possible preference switches: it is unlikely we can make simple predictions without doing this given the vast number of floral factors that affect pollinator preference behaviour (Glover, 2007; Willmer, 2011), or at least understanding better the choice rules that individual pollinators are using. Integrating our understanding of how pollinators are influenced at the local scale with our understanding of their landscape-scale movement is still a greatly unexplored question (Mayer et al., 2011), and there is much scope for combining methodologies such as that presented here with field and laboratory experiments. Similarly, models should also take account of the effects of pollinators on floral resources at the local scale, as this interaction has implications for community ecology and biodiversity (Jeltsch et al., 2013; Qu et al., 2013).

Conclusions

The effects of habitat loss and fragmentation can be explored using a simple spatially-explicit individual-based modelling framework that combines sensible behavioural rules and suitable landscape information; however, suitable rules need to be identified. The approach linking behavioural ecology and landscape ecology that was envisioned by Lima & Zollner (1996) is still relatively unexplored, and we still need to see stronger links between models of movement and landscape ecology, with the results of these models being fed back into experimental manipulations. Ultimately, these can be used to inform conservation strategies (Knowlton & Graham, 2010) and aid our management of the vital pollination services provided by the animals living at the hearts of our agricultural landscapes.

Supplemental Information

Supplemental Information 1 Zipped file containing example C++ code for the model

Three text files are contained, with a detailed explanation for them described in the header of ‘main.cpp’.

Click here for additional data file.

Supplemental Information 2 Sensitivity analysis for the combined effects of pF and number of set-aside fields

Results of sensitivity analysis for the combined effects of pF and number of set-aside fields on the mean and standard deviation of: (A, B) the maximum metric distance travelled from the nest; (C, D) the number of switches between habitat types conducted by the pollinator; and (E, F) the proportion of time the pollinator spends in the wild habitat type. The line colour indicates the value of pF, as detailed in the inset legend.

Click here for additional data file.

Supplemental Information 3 Sensitivity analysis for the combined effects of pS and number of set-aside fields

Results of sensitivity analysis for the combined effects of pS and number of set-aside fields. See legend to Supplemental Information 2 for details. The line colour indicates the value of pS, as detailed in the inset legend.

Click here for additional data file.

Supplemental Information 4 Sensitivity analysis for the combined effects of r and number of set-aside fields

Results of sensitivity analysis for the combined effects of r and number of set-aside fields. See legend to Supplemental Information 2 for details. The line colour indicates the value of r, as detailed in the inset legend.

Click here for additional data file.

Supplemental Information 5 Sensitivity analysis for the combined effects of number of fields initially seeded in the landscape and number of set-aside fields

Results of sensitivity analysis for the combined effects of number of fields initially seeded in the landscape and number of set-aside fields. See legend to Supplemental Information 2 for details. The line colour indicates the number of fields initially seeded, as detailed in the inset legend.

Click here for additional data file.

Supplemental Information 6 Results of sensitivity analysis for the combined effects of pF and hedgerow removal

Results of sensitivity analysis for the combined effects of pF and hedgerow removal. See legend to Supplemental Information 2 for details. The line colour indicates the value of pF, as detailed in the inset legend.

Click here for additional data file.

Supplemental Information 7 Sensitivity analysis for the combined effects of pS and hedgerow removal

Results of sensitivity analysis for the combined effects of pS and hedgerow removal. See legend to Supplemental Information 2 for details. The line colour indicates the value of pS, as detailed in the inset legend.

Click here for additional data file.

Supplemental Information 8 Sensitivity analysis for the combined effects of r and hedgerow removal

Results of sensitivity analysis for the combined effects of r and hedgerow removal. See legend to Supplemental Information 2 for details. The line colour indicates the value of r, as detailed in the inset legend.

Click here for additional data file.

Supplemental Information 9 Sensitivity analysis for the combined effects of number of fields initially seeded in the landscape and hedgerow removal

Results of sensitivity analysis for the combined effects of number of fields initially seeded in the landscape and hedgerow removal. See legend to Supplemental Information 2 for details. The line colour indicates the value of number of fields initially seeded, as detailed in the inset legend.

Click here for additional data file.

Supplemental Information 10 Full post-hoc test results for the results described in Table 1

Click here for additional data file.

Additional Information and Declarations

Competing Interests

Author Contributions

The author declares no competing interests.

Sean A. Rands conceived and designed the experiments, performed the experiments, analyzed the data, contributed reagents/materials/analysis tools, wrote the paper.

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
