# Peer review of "Landscape fragmentation and pollinator movement within agricultural environments: a modelling framework for exploring foraging and movement ecology"

_PeerJ, doi:10.7717/peerj.269_

## Round 0.1 · original submission · Major Revisions

Unfortunately, both the reviewers and I do not feel that your paper is suitable for publication in its current version. The reason for this is that it does not represent a sufficient unit of publication. The paper builds on previous research and attempts to present a formal framework but it does not contain all of the elements that would be expected within a spatially-explicit modeling framework. Reviewer 1 provides a description of the extra elements that would be required if this approach is going to be taken.

Alternatively, if the major goal of the paper is to present the results of the current modeling exercise, then as the author points out, the rules used are too simplistic to be of any real utility. The discussion makes this clear and illustrates some of the requirements necessary to build a useful model. A possible direction for the paper could therefore be to incorporate some of the suggestions made by the author into the model to present a more stand-alone body of work.

Reviewer 1 ·

Basic reporting

This manuscript reports on a modeling framework that can be used to study the spatial movement of animals in a simple landscape. On the one hand the manuscript builds on previous papers by the author (& Whitney) on spatially-explicit modelling by trying to present a more formal framework, while on the other hand the discussion sets up lines of research in which more complex/realistic movement rules are added to the present framework. Whether the inbetween step of this manuscript is required as a stand-alone paper remains to be seen. If the goal is indeed to present a spatially-explicit modelling framework, I would expect all of the following, which are missing now:
- illustrations of generated landscapes and movement pathways
- appendix with all R code used for each of the presented models
- explicit comparison with previously published modelling frameworks by other authors, and a discussion of how the framework presented here adds to and improves those existing frameworks

However, if the goal was to show the findings in figures 1 and 2 as new research findings, it should be explained why these are not as trivial and expected as they seem, as also seems to be concluded by the author himself in the discussion.

minor comments:

fifth line of the abstract: add "the effects of" before "individual behavioural rules"

page numbers would be useful

27: consider naming other factors as well

37: please explain the "help and hinder"

84: this requires more explanation

103: I doubt that terms as Voronoi tessellation and vertex are common knowledge

104: then

112: please explain what example species you have in mind

151: please explain

209: what is the random factor exactly: field vs margin, or simulation runs?

Experimental design

No comments

Validity of the findings

No comments

Reviewer 2 ·

Basic reporting

This piece of modelling simulates the affects of various landscape scenarios on pollinator movement and by inference the quality of the pollination ecosystem service.

Such a process is likely to be scale dependent and its not clear what proportion of pollinator flights operate at the scale simulated. The model pollinators are nest based such as bumble bees rather than hoverflies. There relative significance is not considered.

No real attempt is made to account for natural variation in the parameter values. These are likely to vary through the season with flower density and types, not to mention changes within the bees from queens, first workers and summer workers which all have different behaviour patterns.

There is no sensitivity of the model parameters presented, so we have little feel for the robustness of the model.

I feel (as is so often the case) the value of the pollination ecosystem services claimed in the introduction is overstated.

Several different landscape scenarios are simulated, but relatively little is made of these, certainly in the abstract. It would have been nice to have seen an attempt made to test the model predictions by linking these with measurements (seed set data) from different real life examples.

Experimental design

see above

Validity of the findings

see above

Comments for the author

see above

---

## Round 0.2 · accepted · Accept

Thank you for your re-submission and many apologies for the delay in getting back to you. I was keen to get your manuscript reviewed once more and Christmas and New Year holidays delayed this process. Reviewer 1 is satisfied that the majority of the comments made have been addressed but would still like the modelling approach that you have described compared with other published approaches. I am satisfied however that your re-submitted manuscript is now a sufficient unit of publication, presenting a thorough description of your approach.

Reviewer 1 ·

Basic reporting

The author has dealt with most of comments. More importantly he has clearly chosen for the approach of mainly/merely presenting the modeling framework in this paper. I'm happy that he now includes illustrative landscape illustrations and program code. I also understand that he does not want to turn the manuscript into a review of models, but that's not what I meant. Rather than only presenting your own framework, I expect more comparisons with relevant other published approaches, including a more critical discussion about the novelty and (dis)advantages of the various aspects.

Experimental design

No Comments

Validity of the findings

No Comments

Comments for the author

No Comments